# Incidence Rates of Agricultural Machine-Related Injuries in South Korea

**DOI:** 10.3390/ijerph192315588

**Published:** 2022-11-24

**Authors:** Kyungsu Kim, Hyocher Kim, Minji Lee, Wongeon Jung, Dongphil Choi

**Affiliations:** Agricultural Safety and Health Division, National Institute of Agricultural Sciences, Rural Development Administration, Jeonju-si 54875, Republic of Korea

**Keywords:** agriculture, machine, injury, farmer, occupational

## Abstract

Although agricultural machines are the leading cause of agricultural injury, there are few comparative studies on the injury risks associated with different types of agricultural machines. Therefore, we compared the injury rates and risks of various agricultural machine types in South Korea using data from comprehensive agricultural machine insurance, which is partially paid for by the government. Specifically, we conducted a retrospective cross-sectional study using 2014–2019 data on subscriptions and 2014–2020 data on compensation for personal bodily injury from comprehensive agricultural machine-related insurance coverage. We calculated the agricultural machine-related injury rate for each machine type and analyzed the factors affecting the injury using logistic regression. Between 2014 and 2020, 2061 recorded agricultural machine-related injuries occurred for 338,418 comprehensive agricultural machinery insurance subscriptions. The annual average number of injuries per 1000 agricultural machines was 6.1, showing an annual increase. Multivariate logistic regression analysis revealed that the risk of injury increased with age, which was 2.3 times higher for tillers and power carts than for tractors. There is thus a need for legal safety measures, particularly related to high-risk agricultural machines (e.g., power tillers) and individuals (e.g., older drivers), as well as specific driving licenses and regular inspections of agricultural machinery.

## 1. Introduction

Globally, the occupational mortality and injury rates of farmers have been reported to be higher than those of other occupations [1,2,3,4,5]. Similarly, in South Korea, the rates of mortality and occupational injury among farmers are higher than those among workers in other industries [6,7]. Specifically, machine-related injuries represent the main occupational injuries among farmers [8,9,10], being the main target of agricultural injury prevention. These machine-related injuries accounted for 69.8% of farmers’ deaths in Canada [9] and 67.6% in Japan [10]. In South Korea, they accounted for 53.1% of occupational deaths in 2018 [11] and 31.0% of non-fatal occupational injuries among farmers [12].

Despite prior efforts [9,10,12,13,14,15,16,17,18,19,20] to reveal the magnitude and characteristics of agricultural machine-related injuries, there have been limitations in estimating rates of and comparing the risk of agricultural machine injury by machine type. These studies usually presented the proportions of injuries by machine type and characteristics of injury events. In South Korea, administrative data on agricultural machine-related injuries are produced by some government departments, such as the Korean National Police Agency and the Ministry of the Interior and Safety [14,15]. However, these include simple information, including occurrence registration and the month and place of the occurrence. Furthermore, the administrative data on injuries are compiled based on incidents, making it impossible to calculate the injury rate due to a lack of information about the number of users of the machine or the number of machines used, which form the denominator of the injury rate. The Rural Development Administration conducts a survey with a representative sample on the status and characteristics of occupational injuries, including agricultural machine-related injuries, while also gathering information about the users of the machines [12]. However, it is difficult to obtain reliable results given the insufficient number of injury incidents in this limited sample.

Although these publications provide an image of the size of or the main type of machine for agricultural machine-related injuries, the detailed profile of the injuries in terms of relative risks of injury by machine type and high-risk groups has not been raised. Identifying which agricultural machines are associated with a high risks of injury is essential for setting priorities and targeting for the prevention of agricultural machine-related injuries. In order to perform the exact estimation of machine-specific incidence rate and risk of injuries, information on the users of each machine should be available.

Therefore, the objective of this study was to compare the incidence rate and risk of injuries associated with agricultural machines in South Korea by utilizing agricultural machine insurance data, which contain information on both the subscriptions (users) and compensations (injuries) by machine type to provide basic information to help priority setting for prevention and control of agricultural machine-related injuries.

## 2. Methods

### 2.1. Study Location

The data used in this study were obtained from South Korea. According to the 2020 Census of Agriculture, Forestry, and Fisheries [21], the total number of farms in South Korea is approximately 1.03 million, with a farming population of approximately 2.3 million, or 4.5% of the South Korean population. Farmers cultivate paddy rice (39.6%), fruit trees (16.3%), vegetables (16.3%), and food crops (13.5%). The average cultivation area per farm is 1.08 ha. For rice, 98.6% of the cultivation work is mechanized, whereas for fieldwork, 61.9% is mechanized [22]. Field crops, such as fruit trees and vegetables, utilize smaller fields than rice and still mainly rely on manual labor. Of the farmer population, 42.3% are aged 65 or older, and this population is aging [16]. With the decrease and aging of the farmer population, the dependence on a foreign workforce is increasing.

### 2.2. Data Sources

We used raw data on subscriptions and payments from comprehensive agricultural machine insurance to identify the status of the farmers injured due to agricultural machines. This insurance is operated by the National Agricultural Cooperative Federation (NH Nonghyup), allowing interested farmers to subscribe autonomously. The insurance coverage is subsidized by the state, which pays more than 50% of the insurance fee, following current legal regulations, with additional financial support from local governments for some parts of the fee [23]. This is the only insurance in South Korea that specializes in agricultural machine-related injuries.

The coverage of this insurance consists of four components: compensation for physical injuries sustained by the insured or by other individuals, as well as damage to the insured machine or others’ properties. Most insurance contractors subscribe to all four components. The comprehensive agricultural insurance subscription period is one year, and a contractor must re-subscribe annually [23]. There are 12 types of agricultural machines that can be insured under this system. The insured include the insurance contractors, their families and workers (as individuals over 19 years old), and organizations that own, use, or manage the insured agricultural machine [23]. These data make it possible to estimate the incidence rate of injuries due to agricultural machines by providing information on both the subscription for and payment related to each type of agricultural machine. There were no missing data on either subscriptions or payments.

### 2.3. Analysis Targets

For subscription data, we used the number of individual contracts (338,418 observations) with subscriptions for the coverage of personal bodily injuries under the comprehensive agricultural machine insurance in each year from 2014 to 2019. For payment data, the contracts mentioned above were used to find cases in which the contractors were compensated for their injuries within the 2014–2020 period (2061 cases). As payments of some injury cases occurred after the year of subscription, we traced and extracted compensation payment cases using two years of compensation data for each year’s subscription through a deterministic linkage using the identification number and incident registration number of the subscription contract. The pre-analysis showed that above 96% of the compensation for a year’s subscription was made in that year and the following year; thus, the tracking period of compensation payments could be limited to two years for efficient analysis and generation of the latest statistics. 

### 2.4. Items Analyzed

The data on the subscription to comprehensive agricultural insurance are composed of the contractors’ demographic (gender, age, and region) and insurance (date of subscription, covered items, type of agricultural machine, and subscription unit) data. The data on compensation status include the type of machine that caused the injury, the date of the injury, the date of compensation, and the type of compensation.

### 2.5. Unit for Injury Cases

The unit for the injury cases in this study is an injury event, and each injury was counted based on the incident registration number granted to each individual event based on the comprehensive insurance payment data. For example, if different independent incidents occurred several times under the same contract, different incident registration numbers were given to each event and were recorded in the insurance data, while in the case of several items of compensation within the same incident, a single registration number was assigned.

### 2.6. Statistical Analysis

The number of agricultural machines subscribed and the incidence of injuries was calculated. The incidence rate of machine-related injuries in agricultural workers was indicated as the number of bodily injuries sustained by users per 1000 agricultural machines.

Univariate logistic regression analysis was independently performed to ascertain the association between each risk factor (gender, age, region, and agricultural machinery type) and agricultural machine-related injuries. Multivariate logistic regression analysis was performed with statistically significant predictors through univariate logistic regression and important predictors to build the final prediction model. We used the results to estimate the odds ratio (OR) for agricultural machine-related injuries within a 95% confidence interval (CI). The data were analyzed using the SPSS software, v. 20 (IBM Corporation, Armonk, NY, USA).

## 3. Results

### 3.1. Status of Agricultural Machines in South Korea

According to the statistics published by the South Korean government [24], the most commonly owned and used machines among South Korean farmers are, in the following order, power tillers (usually with a trailer), mini tractors, tractors, and transplanters (Figure 1). In terms of annual trends, the tendency to use power tillers continued to decrease significantly, while the tendency to use tractors increased gradually. 

Among the 12 types of machines that can be insured under the comprehensive agricultural machine insurance plan, tractors accounted for 62.3% of the total number of subscriptions, which was followed by combine harvesters (13.5%), transplanters (10.1%), and power tillers (5.0%). When calculated in connection with the statistics [24] held by the government, the subscription rate (number of subscriptions to agricultural machine insurance/number of agricultural machines in South Korean farms ×100) was 23.8% for tractors, 16.2% for combine harvesters, 8.5% for mini tractors, and 1.3% for power tillers.

### 3.2. Demographic Characteristics of Subscribers Per Agricultural Machine Type

Among the contractors with coverage for personal injuries (338,418) under comprehensive agricultural machine insurance, 96.3% were men and 3.7% were women. Regarding age group, 40.4% of the contractors were in their 60s and 27.3% were in their 50s. For power tillers and power carts, contractors in their 60s or older accounted for 87.1% and 80.0% of users, respectively. For drone sprayers, wide-area chemical sprayers, and balers, contractors under the age of 60 accounted for 78.9%, 67.4%, and 65.0% of users, respectively (Figure 2).

### 3.3. Incidence Rate of Agricultural Machine-Related Injuries and Annual Trends

During 2014–2019, there were a total of 338,418 subscriptions (an average of 56,403 cases per year) by individuals for the coverage of personal bodily injury under comprehensive agricultural machine insurance. Among these, there were 2061 cases of injuries (an average of 344 cases per year). The number of injury incidents per 1000 agricultural machines was 6.1 over the five years, with a tendency to increase annually (5.1 cases/1000 in 2014, 6.7 cases/1000 in 2019). Regarding the type of injury, there was an average of 18.7 injury events per 1000 agricultural machines involving power carts, 18.0 involving power tillers (usually with a trailer), 6.5 involving tractors, and 5.7 involving balers (Table 1). In terms of the yearly trends, the incidence rate of injuries for tractors and combine harvesters tended to increase overall, while the injuries involving power tillers showed an apparent decline.

### 3.4. Injury Rate by Gender, Age, and Region of the Insurance Contract

Overall, the incidence of injuries was higher for women (7.0 cases/1000 machines) than for men (6.1 cases/1000 machines), with different results for each type of agricultural machine. Age-wise, the injury rate among individuals aged 70 years or older (10.0 cases/1000 machines) was 2.6 times higher than those in their 50s (3.9 cases/1000 machines). For most machines, the older the contractor was, the higher the injury rate. This was observed particularly for agricultural power carts and power tillers, where the injury rate was high for those in their 70s or older (Figure 3). Region-wise, the injury rates in Jeollabuk-do and Gyeongsangnam-do were higher than in other regions, and there were different tendencies according to the type of machine. 

### 3.5. Factors Influencing the Incidence Rate of Agricultural Machine-Related Injury

When logistic regression analysis was performed with each variable to examine the effects of gender, age, region, and type of agricultural machine on agricultural worker injuries, we observed that age, region, and type of machine significantly affected the likelihood of an injury (*p* < 0.001). When observing the results of multivariate logistic regression with the independent variables of gender, age, region, and type of agricultural machine, the risk of injury was 1.2, 1.5, and 2.0 times higher for users in their 50s, 60s, and 70s, respectively, compared to those under 50. In terms of the type of agricultural machine, compared to tractors, the risks of injury with power tillers (usually with a trailer) and power carts were 2.3 times higher, while for transplanters and drone sprayers, the risks were 0.1 times higher and for wide-area chemical sprayers, they were 0.2 times higher (Table 2).

## 4. Discussion

This study compared injury rates and the risks related to agricultural machines in South Korea using data from comprehensive agricultural machine insurance. We showed that the risk of agricultural machine-related injuries increased with age and found that the injury rate associated with power tillers and power carts is particularly high.

Agricultural machines have been reported as one of the main sources of occupational injuries, particularly severe and fatal occupational injuries among farmers [9,10,11,12,25]. Among the agriculture-related deaths, machine-related deaths accounted for 69.9% in Canada [9], 67.6% in Japan [10], and 53.1% in South Korea [12]. Agricultural machine-related injuries accounted for 31.0% of non-fatal agricultural injuries in South Korea [12] and for 30.5% of all agricultural injuries in India [16]. Kumar et al. [25] reported that agricultural machine types causing injuries vary by country, but machines with higher energy, such as tractors, seem to be associated with more severe injuries through our literature review [13]. Each country has its own major agricultural products, farming characteristics, agricultural work environments, and average cultivation per farm; thus, the predominantly used machines and injury incidences may vary. According to previous studies, there are differences in the types of major agricultural machines involved in injuries among countries. In the Austrian agricultural and forestry sectors, the incidence of injuries was highest for machines used for timber processing (e.g., chainsaw, disk saw), which was followed by propelled and self-propelled machinery (e.g., tractors) and entrained transport equipment [13]. In India, injuries occurred more frequently in relation to tractors and tractor-operated implements (31%), animal-drawn implements (22%), threshers (14%), and electric motors or pump sets (12%), while tractors were reported to account for 44% of the total occupational deaths among farmers [16]. In Konya, Turkey, tractors accounted for the most frequent cases of injury related to agricultural machines (41.9%), which was followed by trailers and threshers [17]. In the Central Anatolian region of Turkey, tractors, haymakers, and augers accounted, in that order, for most machine-related injuries requiring and upon visits to hospital emergency rooms [18]. In Faisalabad, Pakistan, fodder choppers accounted for most injuries related to agricultural machines (65%), which were followed by rotavators (8.7%), threshers, and sugar cane juice extractors [19].

The results of the present study showed that in South Korea, the most widely used machines are power tillers, which are followed by mini tractors and tractors, among them tillers (usually with a trailer) being the agricultural machines with the highest incidence and risk of injury. The study results showed a similar tendency to the prior studies. According to statistics from the Rural Development Administration in South Korea [12,26], the primary machine type of injury was power tillers, accounting for 35.0–44.0% of non-fatal machine-related injuries, and the other three main machine types of injury were tractors, mini tractors, and brush cutter. Another study [27] reported that power tillers were associated with the highest rate of machine-related injury or damage, which were followed by combines and tractors. The severity of the injuries related to power tillers in South Korean agriculture has been reported in many previous studies [12,20,28,29]. In addition, the main user base of tillers is older farmers [12,29], implying a high risk of injuries.

Power tillers are cheap, multi-functional machines introduced to South Korea in the 1960s through Japan [30] and markedly contributed to the mechanization and improved productivity of South Korean agriculture. Tillers are mainly used for transportation and movement by connecting a trailer to the main body [29]. These tillers with a trailer for transportation have an even lower structural safety, and the protection device for the worker is inadequate, making them leading factors of casualties among farmers. Therefore, injuries with power tillers mainly occurred in relation to transportation (38.6%) in Korea [12]. Injuries related to power tillers and tractors accounted for 61.6% and 11.3% of agricultural machine-related traffic injuries, respectively, in Jeonnam province, Korea [31]. On the other hand, in Japan, tractors accounted for the most frequent cases (46.0%) of machine-related deaths, being tillers as the second most frequent cases (15.1%) during 2004–2013 [10]. In Japan, tillers are usually used for cultivating work without a trailer, unlike in South Korea. Recently, the number of tillers and their injury rate have continued to decrease but, considering that the incidence rate of injury is still high and that there remain a large number of tiller users, prevention measures, such as replacing tillers with safer agricultural machines, are still required.

The results of this study also indicated that while the frequency of injuries is not high for power carts, the injury incidence rate is high, similar to that of power tillers. Tillers have been replaced by power carts, and their use has recently increased; thus, the high injury rate, despite power carts being a new, safer machine, requires attention. This is likely a result of various factors, including inadequate safety systems for agricultural machines, older farmers as the main users, the poor condition of farm roads, and frequent use as the main means of carrying and transportation, all of which are the same risk factors as for injury events involving power tillers. 

In particular, agricultural machinery injury during transportation needs to be concerned. Lee et al. [32] reported that the main cause of injury deaths was transport accidents (41.8%) among workers in agriculture, forestry, and fishing, while they also found intentional self-harm (50.7%) among workers in all industries using death registration data in South Korea. Agricultural machine-related traffic injuries had higher rates of fatality than all traffic injuries by more than seven times in South Korea [33] and by about five times in the United States [34]. Among nonfatal agricultural injuries treated in emergency departments in the United States, the most common event of injury was transportation (33.9%), and the main sources of injuries were vehicles (mainly tractor) [35].

In South Korea, driving licenses, vehicle registration, and regular inspection are legal obligations for automobiles in general. However, there is no safety management for agricultural machines, except for the legal safety test standards for agricultural machines to be produced and distributed. Consequently, there is a need to draw attention to the safety measures for agricultural machines that are widely used by older individuals for carrying and transportation at farming sites.

There is also a need for the overlapping application of various measures, including improvement of the technical safety of high-risk machines, replacement of existing high-risk machines with safer alternatives, and mandating safety systems such as driving licenses, regular safety inspections, and safety training for using the machines. Furthermore, there is a need to improve the safety of the roads and working environments, promote safety management awareness, and educate users on safety behaviors. To strengthen the technical safety of agricultural machine-related injuries, technical measures such as attachment of tractor roll over protection structures [36], injury detection and injury prevention systems [37,38,39], and advanced technologies such as ICT and sensors may be used.

There are several limitations of the present study. The main limitation of this study is that agricultural machine insurance is optional, and there is a limit to generalizing the situation to the entire agricultural machinery use in South Korea because of the low subscription rate, particularly for power tillers, which are owned by many farmers and are frequently used. In addition, there is a lack of detailed information about the occurrence of injuries, such as places, types of work operation, and types of injury, as these data mainly contain the information required to elicit insurance payment. There is a need to improve compensation data by collecting more detailed information on the occurrences of injuries (i.e., through revisions in the application forms for insurance money). Finally, there is a limitation regarding methodological approaches in analysis. Future studies are recommended to analyze data using advanced methodological approaches (e.g., survival analysis, decision tree analysis, and so on) to drive more rigorous and practical implications. Despite these limitations, our study used empirical data on insurance subscriptions and payments, and the data had undergone on-site investigations of damage and various public data (e.g., police records, medical records, and death registration data) for every case, adding to the reliability of our findings.

As for the strength of the present study, this study utilized large-scale nationwide data (more than 300,000 cases) from the only insurance in South Korea that specialized in agricultural machine-related injuries and used the information on both the subscription for and payment related to each type of agricultural machine. With these advantages, this study was able to estimate user-based valid incidence rates by the machine type and to compare the risk of injuries by the machine type, which provides basic information to help make effective prevention strategies on priority setting for high-risk groups.

## 5. Conclusions

This study compared the incidence and risk of injuries by agricultural machinery by utilizing a comprehensive dataset of agricultural machine insurance provided by the South Korean government to provide basic evidence for the prevention of agricultural machine-related injuries among farmers. The number of injuries per 1000 agricultural machines was 6.1 annually. Most injuries involved power carts and tillers (usually with a trailer), which is followed by tractors and balers. The results of multivariate logistic regression analysis showed that the risk of an injury was 1.0, 1.5, and 2.0 times higher for individuals in their 50s, 60s, and 70s, respectively, than for those under 50. Regarding the type of agricultural machinery, the risk of injury was 2.3 times higher for power tillers and agricultural power carts than for tractors. Our study thus emphasized the urgent need for injury prevention measures for older individuals using agricultural machines, for special attention for users of high-risk agricultural machines (e.g., tillers, power carts), and for legal safety management measures such as mandating agricultural machinery driving licenses and regular machine inspection. Future research is required to investigate the causes and characteristics of injuries for practical prevention measures, particularly injuries related to high-risk agricultural machines.

## Figures and Tables

**Figure 1 ijerph-19-15588-f001:**
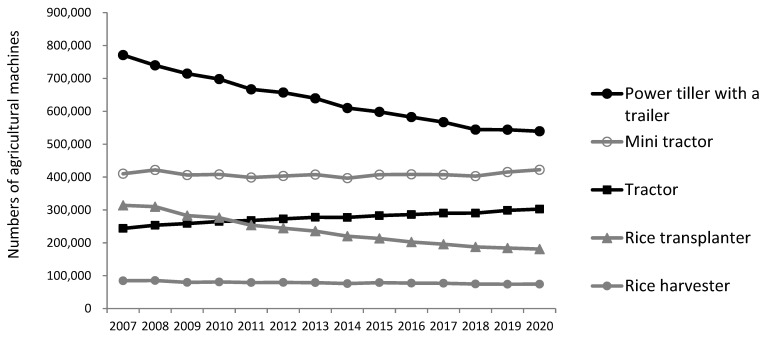
Number of agricultural machines in South Korea.

**Figure 2 ijerph-19-15588-f002:**
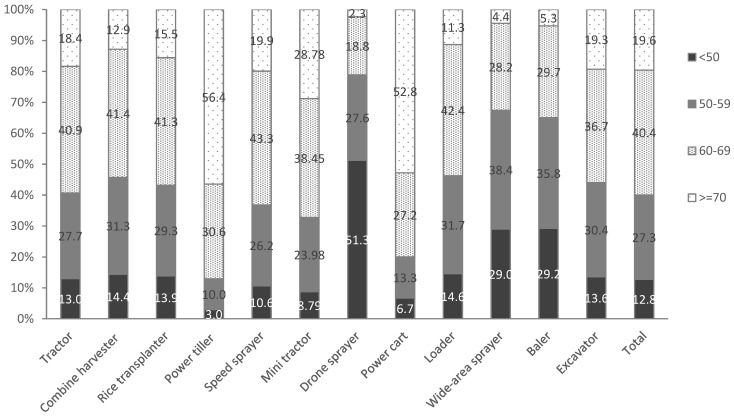
Ages of agricultural machine insurance contractors by agricultural machinery type.

**Figure 3 ijerph-19-15588-f003:**
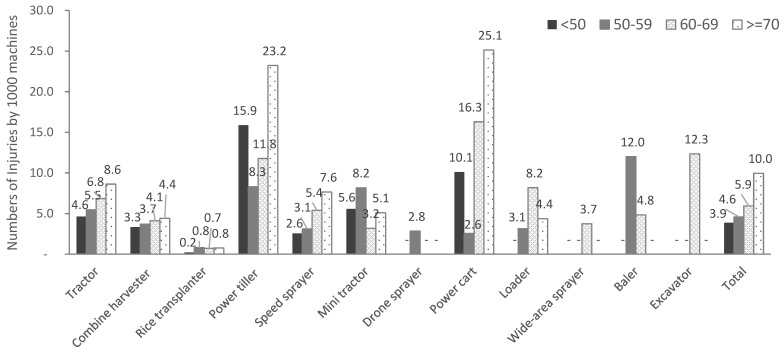
Incidence of injuries related to machines by type of machine and age among subscribers from 2014 to 2019.

**Table 1 ijerph-19-15588-t001:** Incidence rate of injuries related to agricultural machines among subscribers from 2014 to 2019.

Categories	Tractor	Combine Harvester	Rice Transplanter	Power Tiller	Speed Sprayer	Mini Tractor	Drone Sprayer	Power Cart	Loader	Wide-Area Sprayer	Baler	Excavator	Total
Subscription (N)	210,742	45,845	34,077	16,938	18,406	4086.00	1278	2939	2020	948	698	441	338,418
Compensation (N)	1369	179	23	305	91	21.00	1	55	10	1	4	2	2061
Injury rate *	6.50	3.90	0.67	18.01	4.94	5.14	0.78	18.71	4.95	1.05	5.73	4.54	6.09

* Number of injury cases per 1000 machines.

**Table 2 ijerph-19-15588-t002:** Odds ratio (by logistic regression analysis) of the determinants of the occurrence of injuries related to agricultural machinery.

Variables	Categories	Crude ORs (95% CI)	Adjusted ^(a)^ ORs (95% CI)
Gender	Males	1	1
	Females	1.13 (0.91–1.40)	1.06 (0.85–1.32)
Age (years)	<50	1	1
	50~59	1.21 (1.01–1.46) *	1.22 (1.02–1.47) *
	60~69	1.55 (1.31–1.84) ***	1.50 (1.27–1.78) ***
	≥70	2.60 (2.19–3.10) ***	2.04 (1.71–2.44) ***
Region	Jeju-do	1	1
	Gyeonggi-do	1.44 (0.74–2.79)	1.67 (0.86–3.24)
	Gangwon-do	1.59 (0.81–3.12)	1.79 (0.91–3.52)
	Chungchoeongbuk-do	1.61 (0.82–3.15)	1.83 (0.93–3.58)
	Chungchongnam-do	1.80 (0.93–3.49)	2.12 (1.10–4.12) *
	Jeollabuk-do	2.15 (1.11–4.19) *	2.63 (1.35–5.12) **
	Jeollanam-do	1.85 (0.95–3.60)	2.40 (1.23–4.68) *
	Gyeongsangbuk-do	1.70 (0.87–3.30)	2.12 (1.09–4.13) *
	Gyeongsangnam-do	2.09 (1.07–4.11) *	2.11 (1.07–4.14) *
Agricultural machine	Tractor	1	1
	Combine harvester	0.60 (0.51–0.70) ***	0.60 (0.51–0.70) ***
	Rice transplanter	0.10 (0.07–0.15) ***	0.10 (0.07–0.15) ***
	Power tiller	2.76 (2.43–3.14) ***	2.29 (2.00–2.62) ***
	Speed sprayer	0.75 (0.61–0.93) *	0.72 (0.57–0.90) **
	Mini tractor	0.77 (0.49–1.20)	0.74 (0.47–1.15)
	Drone sprayer	0.12 (0.02–0.87) *	0.14 (0.02–0.97) *
	Power cart	2.76 (2.08–3.66) ***	2.32 (1.74–3.09) ***
	Loader	0.70 (0.36–1.35)	0.76 (0.40–1.47)
	Wide-area sprayer	0.17 (0.02–1.18)	0.17 (0.02–1.24)
	Baler	0.90 (0.34–2.41)	1.02 (0.38–2.73)
	Excavator	0.71 (0.18–2.86)	0.75 (0.19–3.01)

Abbreviations: OR, odds ratios; 95% CI, 95% confidence interval; ^(a)^ Adjusted for gender, age, region and agricultural machine; * *p* < 0.05, ** *p* < 0.01, *** *p* < 0.001.

## Data Availability

Data may be obtained from a third party and are not publicly available.

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
