# Peer review of "Incidence Rates of Agricultural Machine-Related Injuries in South Korea"

_ijerph, 2022, doi:10.3390/ijerph192315588_

Round 1

Reviewer 1 Report

An interesting manuscript with valuable information and results! It gives a deeper understand of the challenges with agricultural machine-related injuries in South Korea.

A map would have been valuable to get an understanding of regions discussed. 

The results would have been even more interesting if the location of injury was presented: road, field, farm yard ... and during what type of work operation as well as the consequenses for the farmer ...type of injury to the body and...  However ...this kind of information seems to be lacking in the original source of data. 

A good discussion, but it might be of interest with Conclusions instead of a summary in the end?

Reviewer 2 Report

This study analyses the incidence and injury risks associated with different types of agricultural machines in South Korea using data collected from the insurance company. The factors affecting the injury are evaluated using logistic regression. The main limitation of the study is the considered factors for the analysis. This is mainly attributed to the scarcity of data. The depth of the analysis is also shallow. However, the authors clearly clarify this limitation, and the study findings are well presented despite the limited parameters in the dataset. The results and conclusions are contributory and have practical implications.  

 Few more comments:

The abstract is well written, with all the significant points.

There is a missing gap between the last paragraph and the previous paragraph. Better to use some connecting sentences on the data used for this study and the merits of this data.

A few pieces of literature are cited at the very beginning of the introduction. But there is no comprehensive review of the relevant studies. If there are no local studies, there are plenty of international studies. Please incorporate some of those and justify your contribution.

Table 2: Use p-value with ORs. It will provide a vivant understanding of the level of significance of the factors.

The main strength of the paper is the discussion. It is well described. Limitations are also described but mainly from the data point of view. As mentioned, from the methodological perspective, the study is very shallow, particularly using regression derivatives, i.e., logistic regression, an ancient, simple, and standard method. Better to admit this as a limitation and suggest advanced methodological approaches, e.g., using advanced multivariate regression techniques, as the further research direction. 

Reviewer 3 Report

1-     In line 40, the authors stated that “In South Korea, they accounted for 56.9% of occupational deaths”. In my opinion this statement is not realistic, when I checked its reference (reference No. 10), I found that this reference is not true, or incorrectly referenced. Please explain about this statement? 

2-     The authors need to explain the novelty of their work in prevention of machine-related accidents, in the “Introduction” section

3-     In section “2.2. Data sources” line 84, it is recommended that the data span for payments and accidents to be defined in analogous manner, i.e. 2014 to 2019 or 2014 to 2020

4-     In line 86, the explanation of “comprehensive agricultural machine insurance” is not necessary or must be minimized.

5-     In my opinion, “Figure 1” is not necessary and can be deleted.

6-     In section “2.6. Statistical analysis”, line 136, the authors required to clarify the statement “Technical statistics for the number of agricultural machines and the incidence of injuries were calculated”. Are the number of agricultural machines and the incidence of injuries, the “technical statistics”? . It is required that explain this statement.

7-     In section “2.6. Statistical analysis”, what is the rationale for selection of the “univariate and multi-variate logistic regressions” to conduct statistical analysis? Please explain.

8-     “Figure 2.” need to be refined. Please display the vertical axes of the chart

9-     One of the required sections for every scientific article is the “Conclusion” section. While in the current article, I did not see it. Please add this section to the article.
